TECHNICAL RELEASE

# Efficiently constructing complete genomes with CycloneSEQ to fill gaps in bacterial draft assemblies

Hewei Liang[1,2,3,†], Yuanqiang Zou[3,4,†], Mengmeng Wang[1,5], Tongyuan Hu[1,2], Haoyu Wang[1,5], Wenxin He[1], Yanmei Ju[2], Ruijin Guo[2], Junyi Chen[1,6], Fei Guo[1,6], Tao Zeng[1,6], Yuliang Dong[1,6], Yuning Zhang[1,6], Bo Wang[4,7,8], Chuanyu Liu[1], Xin Jin[1], Wenwei Zhang[1], Xun Xu[4,*] and Liang Xiao[3,4,*]

1  BGI Research, Shenzhen 518083, China
2  BGI Research, Wuhan 430074, China
3  Shenzhen Engineering Laboratory of Detection and Intervention of Human Intestinal Microbiome, BGI Research, Shenzhen 518083, China
4  State Key Laboratory of Genome and Multi-omics Technologies, BGI Research, Shenzhen 518083, China
5  College of Life Sciences, University of Chinese Academy of Sciences, Beijing 100049, China
6  BGI Hangzhou CycloneSEQ Technology Co., Ltd, Hangzhou 310030, China
7  China National GeneBank, BGI Research, Shenzhen 518120, China
8  Shenzhen Key Laboratory of Environmental Microbial Genomics and Application, BGI Research, Shenzhen 518083, China

**Submitted:**  04 November 2024

\* Corresponding authors. E-mail: xuxun@genomics.cn; xiaoliang@genomics.cn

† Contributed equally.

Preprint submitted at https://doi.org/10.1101/2024.09.05.611410

## ABSTRACT

Current microbial sequencing relies on short-read platforms like Illumina and DNBSEQ, which are cost-effective and accurate but often produce fragmented draft genomes. Here, we used CycloneSEQ for long-read sequencing of ATCC BAA-835, producing long-reads with an average length of 11.6 kbp and an average quality score of 14.4. Hybrid assembly with short-reads data resulted in an error rate of only 0.04 mismatches and 0.08 indels per 100 kbp compared to the reference genome. This method, validated across nine species, successfully assembled complete circular genomes. Hybrid assembly significantly enhances genome completeness by using long-reads to fill gaps and accurately assembling multi-copy rRNA genes, unlike short-reads alone. Data subsampling showed that combining over 500 Mbp of short-read data with 100 Mbp of long-read data yields high-quality circular assemblies. CycloneSEQ long-reads improves the assembly of circular complete genomes from mixed microbial communities; however, its base quality needs improving. Integrating DNBSEQ short-reads improved accuracy, resulting in complete and accurate assemblies.

**Subjects**  Genetics and Genomics, Microbiology, Bioinformatics

## BACKGROUND

Current microbial sequencing is primarily based on short-read sequencing technologies [1], including mainstream platforms such as Illumina and DNBSEQ, for both isolated genome and metagenomic studies [2–4]. Short reads are favored for their low cost and high accuracy [5]. However, assemblies based on short reads typically result in a draft genome [6], which is presented as a few to several hundred contigs. Fragmented assemblies hinder the comprehensive understanding of bacterial functions [7, 8]. In this decade, short-read sequencing has been widely applied in the microbial field, contributing to a

large number of diverse draft genomes [2–4]. However, it remains challenging to close the gaps between contigs in these draft genomes by relying only on short reads.

Long-read sequencing has been developed for nearly two decades [9]. By using long-read assemblies, or hybrid assemblies combining long reads with short reads, the genome completeness and proportion of complete circular genomes assembled can be significantly increased [10]. Despite this, the cost of long-read sequencing remains significantly higher than that of short reads [11]. It limits the widespread application of long-read sequencing, especially in large-scale datasets. Currently, there are only a few self-produced large-scale datasets, such as the NCTC3000 [12] and the *Actinomycete* genomes datasets [13]. Complete bacterial genomes can provide comprehensive insights into genomic structures, promote the identification of novel genes, and enhance our understanding of microbial evolution [14].

CycloneSEQ is a newly developed long-read sequencing platform developed by BGI-Research using novel nanopore technology to perform long-read sequencing [15]. It has demonstrated excellent performance in sequencing the *Escherichia coli* genome. However, its performance in sequencing diverse microbial genomes has not yet been systematically evaluated.

This study focused on assessing the performance of CycloneSEQ in microbial sequencing and the improvements in genome assembly achieved using CycloneSEQ long-read. By integrating short reads from DNBSEQ with long reads from CycloneSEQ in a hybrid assembly, we validated the effectiveness of this approach in assembling complete circular genomes. Complete genomes enhance our understanding of functional gene coding in bacteria. Additionally, we explored the data volumes required for assembly. By performing random subsampling of the long-read and short-read sequencing data, we tested the assembly performance within data ranges of 100 Mbp, 200 Mbp, 500 Mbp, and 1000 Mbp, providing insights into the success rate of achieving complete assemblies and their accuracy at different data volumes.

## RESULTS

### Sequencing and genome assembly for ATCC BAA-835

In order to evaluate the accuracy of CycloneSEQ sequencing and the quality of the assembled genome, we cultured the commercial standard strain ATCC BAA-835 of *Akkermansia muciniphila* and performed CycloneSEQ and DNBSEQ sequencing on its extracted DNA. We obtained high-depth sequencing data with both short-read and long-read sequencing exceeding 1000× coverage, including 12.07 Gbp of long-reads with an average length of 11,659.2 bp (Figure 1A), and paired short-reads for a total of 4.10 Gbp and an average length of 99.9 bp (Table 1). The average quality of the long reads was 14.4 (Figure 1B), which improved to 14.9 after quality control by selecting reads with a quality value greater than Q10. The paired short-reads had an average quality of 35.8 and 34.9, respectively (Table 1).

As the use of different sequencing reads and assembly methods can affect the final assembly results, we chose the widely-used software Unicycler [16] (which relies on SPAdes [17] for short-read assemblies) and Flye (RRID:SCR_017016) [18] to perform assemblies using only short-reads, only long-reads, and a hybrid of both.

Both long-reads and hybrid assemblies resulted in a single circular genome, while the short-read assembly resulted in 46 contigs (Figure 2A). The GC content was slightly affected

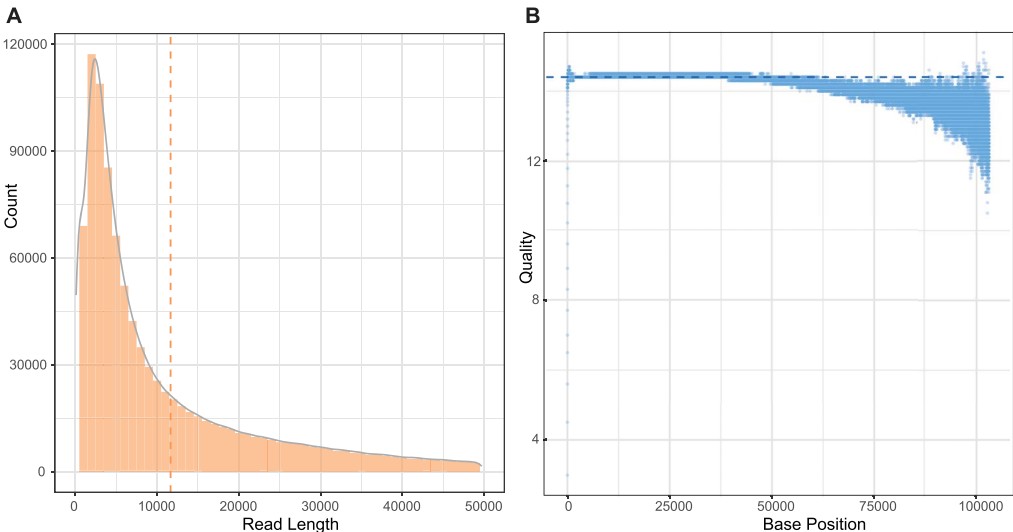

**Figure 1.** **Raw data information from the sequencing.**
(A) The bar plot denotes the count of reads in different length ranges, and the curve line denotes the density of read lengths. (B) The quality of each base position in each read.

**Table 1.** Information on the sequencing data of the type strain.

| File | Sequencing platform | Seqs count | Bases | Average length |
|---|---|---|---|---|
| ATCC-longread | CycloneSEQ | 697,978 | 9,347,774,629 | 13,392.60 |
| ATCC-shortread_1 | DNBSEQ | 41,061,691 | 4,101,831,871 | 99.9 |
| ATCC-shortread_2 | DNBSEQ | 41,061,691 | 4,101,518,531 | 99.9 |

by the completeness and accuracy of the assembly, with the short-read assembly being 55.74%, the long-read assembly being 55.75%, and the hybrid assembly being consistent with the reference at 55.76% (Figure 2B). In terms of total length, the hybrid assembly's length of 2,664,100 bp was closest to the reference's 2,664,102 bp, while the long-read assembly was 2,661,711 bp and the short-read assembly was 2,635,075 bp (Figure 2C), indicating that the short-read assembly had much more fragmentary gaps.

As for the error rate, the short-reads achieved a quality of Q35. Thus, the short-read assembly exhibited only 0.04 mismatches and 0.11 indels per 100 kbp (Figure 3). The hybrid assembly, which was based on the short-read assembly, had 0.08 mismatches and 0.15 indels per 100 kbp. By contrast, the long-read assembly's error rate was several hundredfold higher, with 13.53 mismatches and 127.49 indels per 100 kbp (Figure 4). Such a high error rate could badly affect subsequent analyses of the assembly. Overall, we considered the hybrid assembly to be the optimal assembly method.

## Hybrid assembly enhances genome accuracy and completeness

To evaluate the performance of CycloneSEQ and DNBSEQ on actual samples, we collected 10 strains from 9 diverse species for sequencing. The long-read sequencing data for these 10 strains had average quality scores ranging from 12.3 to 15.5 (Table 2). We then assembled the data using only short reads, only long reads, and a hybrid of both. The hybrid assemblies consistently resulted in circular genomes (Figure 4), including potential small circular genomes from bacteriophages or plasmids. With long-read assemblies,

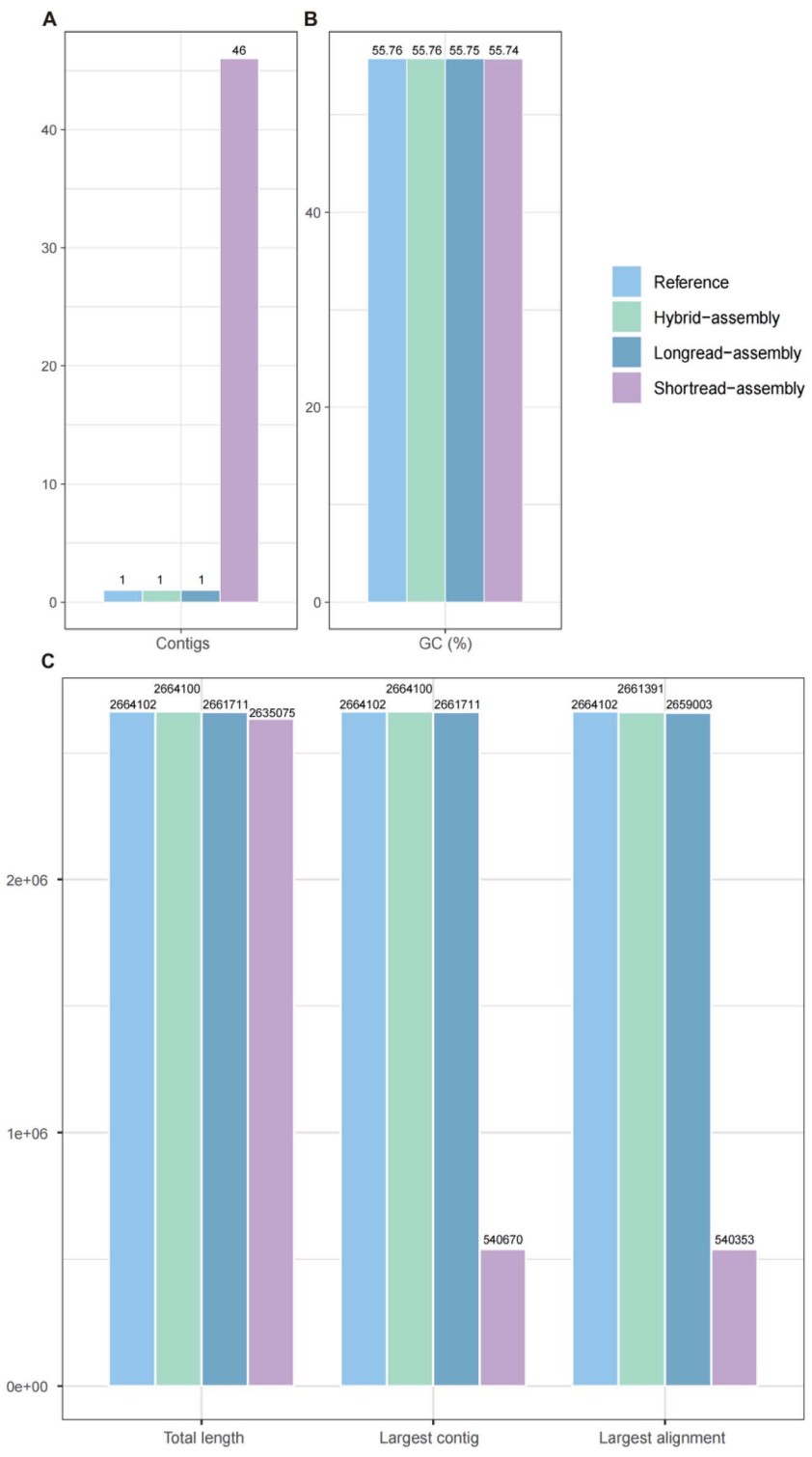

**Figure 2.** Evaluation of the type strain genome using short-read assembly, long-read assembly, and hybrid assembly.
(A) Number of contigs. (B) GC content. (C) Total length, length of largest contig and largest alignment (bp) of the assembled genome.

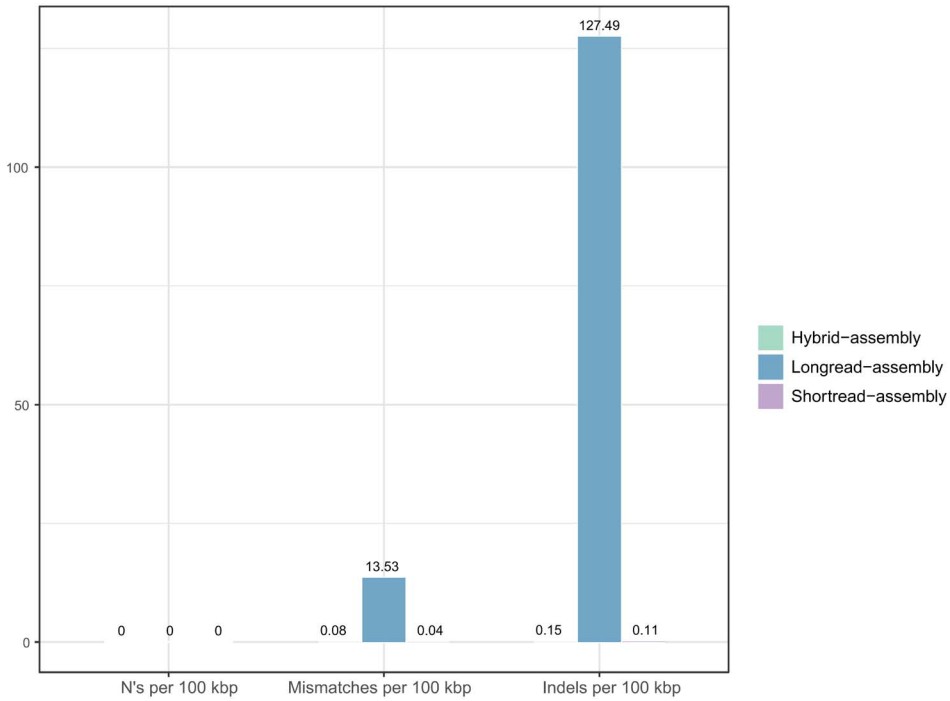

**Figure 3.** **The differences in counts of N's, mismatches, and indels per 100 kbp for ATCC BAA-835 between reference genome and the assembled genome among short-read, long-read, and hybrid assemblies.**

**Table 2.** Overview of the long-read sequencing data.

| Sample | Average_len | Bases | %A | %C | %G | %T | %N | avgQ |
|---|---|---|---|---|---|---|---|---|
| ATCC_BAA835 | 11,659.2 | 12,070,135,504 | 22.3 | 27.7 | 27.6 | 22.4 | 0 | 14.4 |
| AM114-1B | 6,335.07 | 9,710,736,189 | 34.2 | 18.1 | 16.9 | 30.8 | 0 | 15.5 |
| AM114-28 | 7,051.22 | 7,064,400,097 | 35.8 | 17.1 | 15.8 | 31.3 | 0 | 13.3 |
| AM114-5B | 8,210.16 | 8,437,886,665 | 24.3 | 28.9 | 27.5 | 19.3 | 0 | 12.4 |
| AM114-19B | 9,921.59 | 5,315,394,125 | 39.8 | 22.6 | 20.5 | 17.1 | 0 | 12.3 |
| AM114-25B | 7,433.67 | 11,942,285,762 | 31.7 | 20.7 | 19.6 | 28 | 0 | 13.8 |
| AM114-17B | 4,247.93 | 2,087,988,461 | 30.1 | 21.8 | 20.4 | 27.7 | 0 | 15.5 |
| AM114-O-1 | 6,613.17 | 9,181,278,316 | 26.1 | 25.3 | 24.4 | 24.1 | 0 | 14.3 |
| AM114-27B | 9,349.46 | 6,054,785,494 | 32.6 | 20.2 | 18.9 | 28.3 | 0 | 13.8 |
| AM114-O-11 | 10,411.5 | 2,649,028,894 | 46.7 | 15.1 | 14 | 24.2 | 0 | 13.1 |
| AM114-O-9 | 5,040.47 | 5,816,251,540 | 26.9 | 25.2 | 23.9 | 24 | 0 | 15.4 |

8 out of 10 strains were successfully assembled into circular genomes, whereas short-read assemblies did not result in circular genomes. When compared to the hybrid assembly genomes, the long-read assemblies exhibited more than 27.97 indels per 100 kbp, while the short-read assemblies showed almost no variation, with fewer than 0.18 indels per 100 kbp (Figure 5). These findings are similar to those from ATCC BAA-835 analyses, where long-read assemblies tend to be more error-based, and short-read assemblies are fragmented.

For these 10 test samples, we further analyzed the circular genomes by hybrid assembly. According to the GTDB taxonomic annotation (RRID:SCR_019136) [19], they could be classified into five phyla, five classes, six orders, six families, eight genera, and nine species, which included two strains of *Escherichia coli* (Figure 4). The GC content of these strains

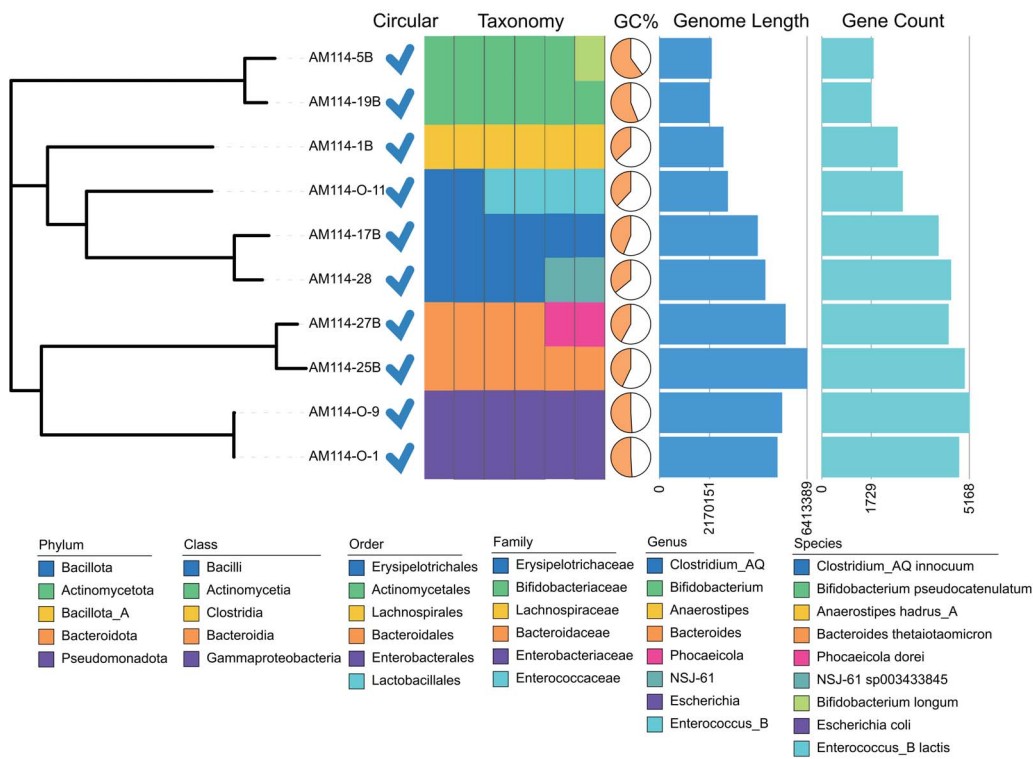

**Figure 4. The phylogenetic tree of the 10 strains.**
All the genomes are circular. Different taxonomic levels and classification information are indicated with relevant colors. In the pie chart, the orange coverage represents the GC content. The length and number of genes for each genome are indicated by bar plots.

varied from 36% to 60%. The size of the genomes ranged from 2.17 Mbp to 6.41 Mbp, and the gene counts ranged from 1,729 to 5,168. These assemblies suggest that the hybrid assembly approach, integrating DNBSEQ short-reads and CycloneSEQ long-reads, capitalizes on the strengths of both long- and short-reads to assemble complete and accurate genome assemblies for common types of bacteria.

## Long-reads restore multi-copy genes by filling gaps

Hybrid assembly effectively enhances genome completeness, and the improvements brought by long-reads to the draft assembly are noteworthy. Evaluating these ten diverse genomes from the perspective of basic functional elements, the complete genome shows a significant increase in the number of coding sequence (CDS), rRNA, and tRNA coding genes compared to the draft (Figure 6A). The increase in the number of rRNAs is particularly notable, including 5S rRNA, 16S rRNA, and 23S rRNA. In particular, 5S rRNA, 16S rRNA, and 23S rRNA often appear as a cluster close to 4,500 bp in length in the genome (5S rRNA: 68-111bp, 16S rRNA: 1519-1558bp, 23S rRNA: 2869-3056bp) (Figure 7A), and they appear as multiple copies at different locations in the genome map (Figure 6B), thus adding to the challenges of short-read assemblies. In the draft genome assembled using short-reads, there is often only one set of rRNA, while other regions containing rRNAs become gaps.

By mapping short-reads to the complete genome, all the base sites of the AM114-1B, AM114-5B, and AM114-19B genomes had a coverage depth of more than 100× (Figure 6C).

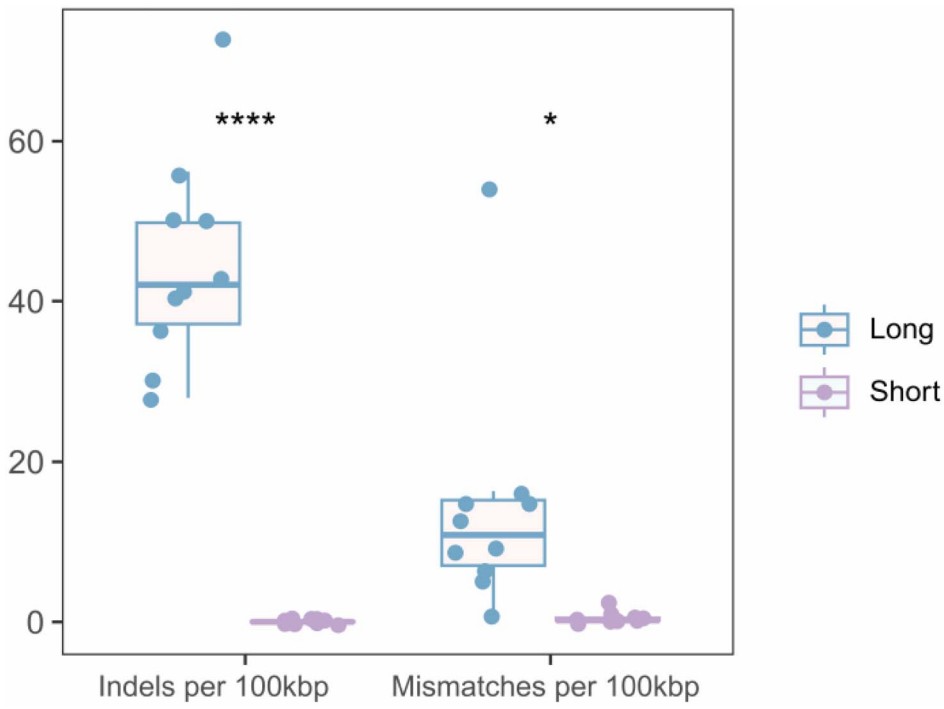

**Figure 5.** The error rate of short-read and long-read assemblies compared to hybrid assemblies.

However, gap regions persist in the short-read assemblies. In other complete genomes, continuous sites with depths fewer than 100× were found (Figure 6C), indicating that both the lack of short-read sequencing depth and multicopy regions contribute to the gap regions in short-read assemblies. When long reads were mapped to the complete genome, there were reads longer than 5 kbp that could cover the gap regions in the draft, acting as a bridge and guiding genome assembly, thereby making the connections possible.

## Optimal data volumes for complete circular genome assembly

In order to determine the minimum data volume required for assembling complete circular genomes, we randomly generated five repetitions of the long-read and short-read sequencing data from 10 actual samples, with subset sizes of 100 Mbp, 200 Mbp, 500 Mbp, and 1,000 Mbp each. This resulted in a total of 400 subsets (5 replicates ∗ 4 subset sizes ∗ 2 read types ∗ 10 samples). Subsequently, we permuted these subsets from the same sample and assembled them.

When 1,000 Mbp of short-read data was combined with either 1,000 Mbp or 500 Mbp of long-read data, 9 out of 10 or 8 out of 10 samples, respectively, achieved 100% (5/5) assembly into circular genomes (Figure 8). Remarkably, even with 200 Mbp or 100 Mbp of long-read data, respectively, 7 out of 10 or 5 out of 10 samples achieved complete assembly into circular genomes in all five replicates. Overall, when using 1,000 Mbp of short-read data as a base and combining it with 100 Mbp, 200 Mbp, 500 Mbp, and 1,000 Mbp of long-read, the rates of achieving circular complete genomes were 76%, 88%, 94%, and 96%, respectively. This result suggests that, for the assembly of bacterial genomes of common

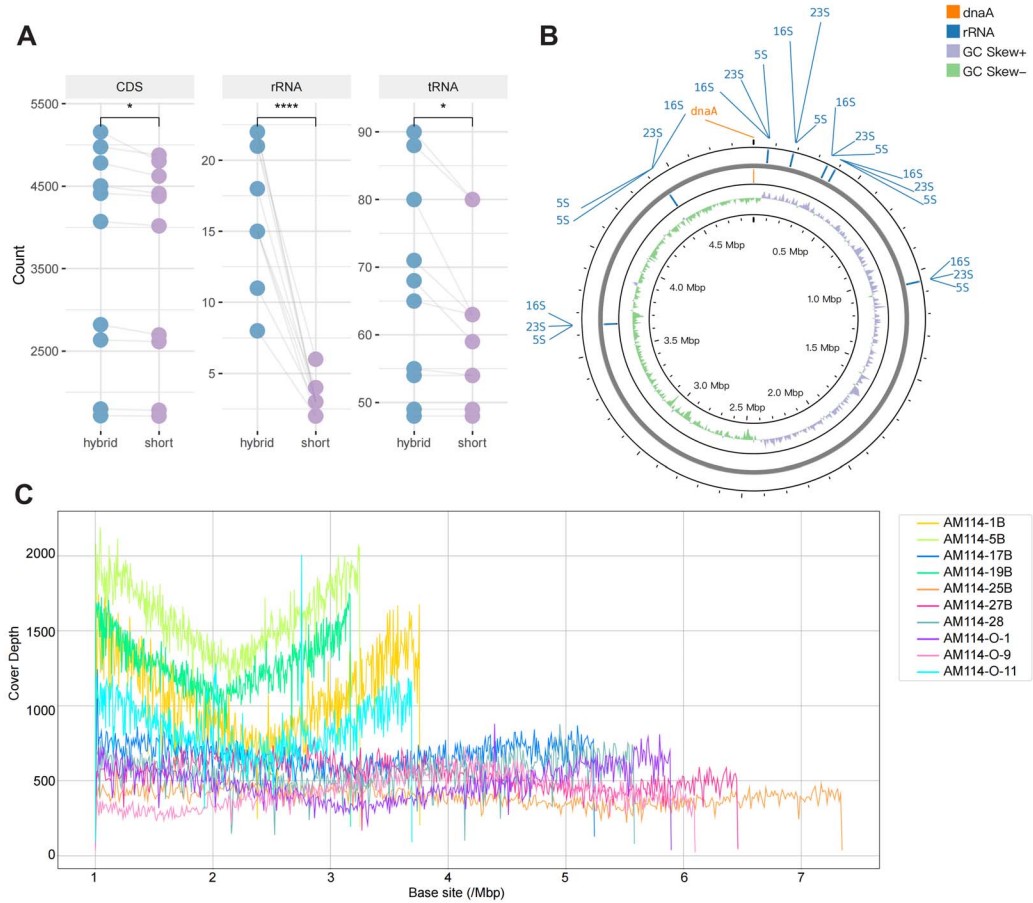

**Figure 6. Comparison of complete genomes and short-read assemblies.**
(A) The gene count annotated as coding sequences (CDS), ribosomal RNA (rRNA), and transfer RNA (tRNA). The complete genomes and short-read assemblies of the same strain were linked, and the paired values were compared using the Wilcoxon test. (B) The circos plot of the chromosome genome of AM114-O-1, with all 22 rRNA positions indicated on the graph. (C) Covering depth of each base position on the complete genome by short-read sequencing.

species and sizes, using 1,000 Mbp of short-read data combined with more than 100 Mbp of long-read data greatly increases the likelihood of achieving a complete genome assembly.

Hybrid assemblies using 500 Mbp of short-read data were slightly inferior to those using 1,000 Mbp of short-read data for the strains. However, overall, they still achieved complete genome assembly at rates of 74%, 84%, 84%, and 88%, respectively. When the volume of short-read data was 200 Mbp, the rates of complete genome assembly were only around 50%; even when combined with 1000 Mbp of long-read data, only 3 out of 10 strains were fully assembled into circular genomes across all five replicates (Figure 8). Also, when using only 100 Mbp of short-read data, the rates of complete genome dropped markedly, resulting in an overall rate of only 34.5% (see rates of complete genome table in GigaDB [20]). It is noteworthy that in assembly approaches reliant on short-read, the volume of short-read data is crucial for achieving a complete assembly. When there is an adequate volume of short-read data, just a few hundred Mbp of long-read data can suffice to produce an excellent complete assembly.

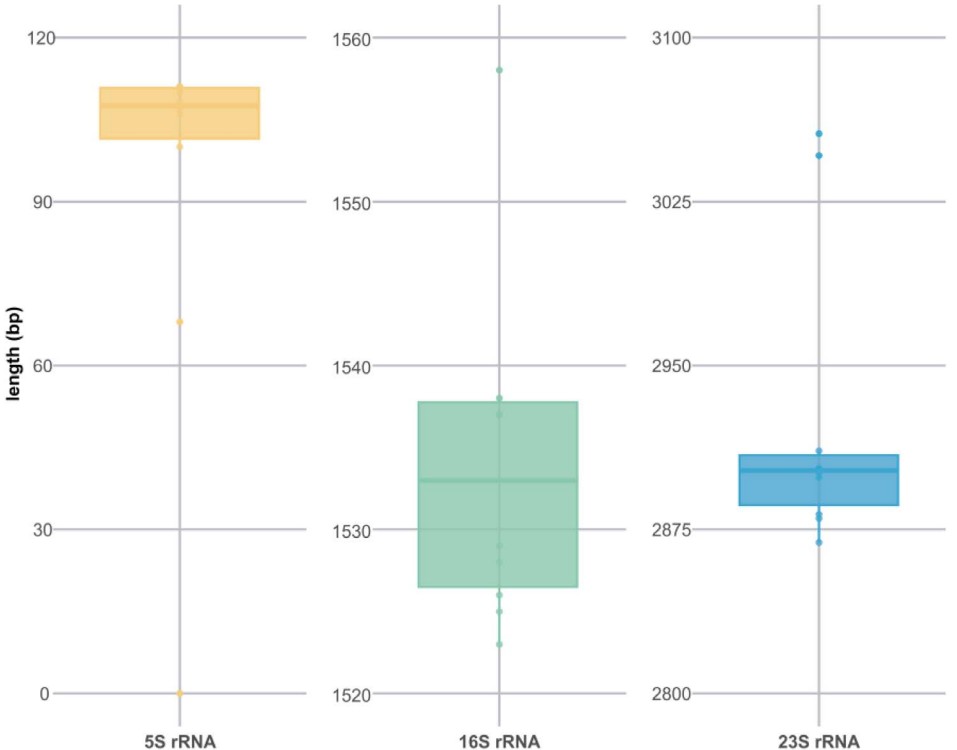

**Figure 7.** The lengths of 5S rRNA, 16S rRNA, and 23S rRNA of ten test strains using hybrid assembly.

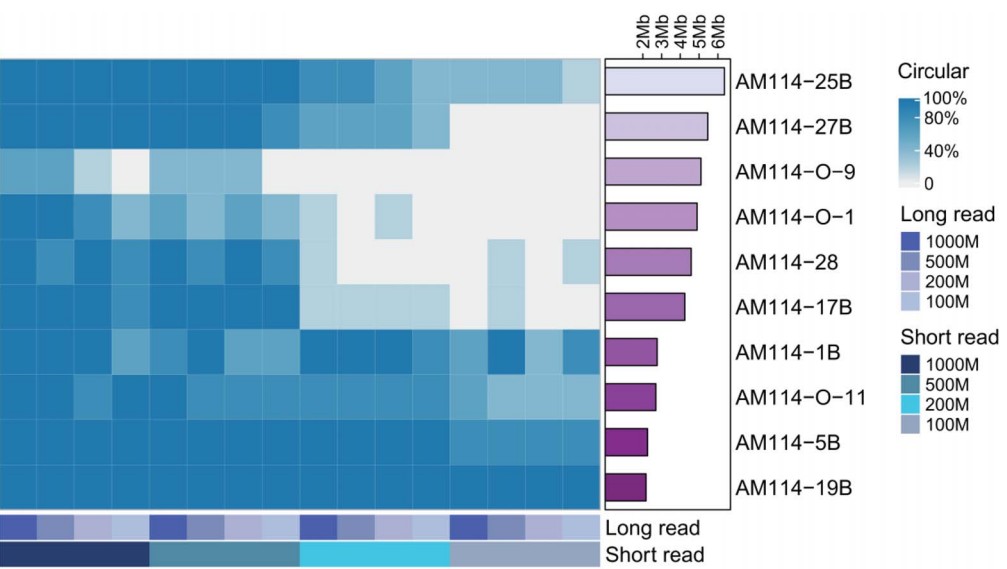

**Figure 8.** The proportion of circular genomes assembled under different data volumes.
For each strain, 5 sets of data with volumes of 100Mb, 200Mb, 500Mb, and 1000Mb are randomly sampled for hybrid assembly. The proportion is calculated based on the number of circular chromosomes formed in the 5 genomes.

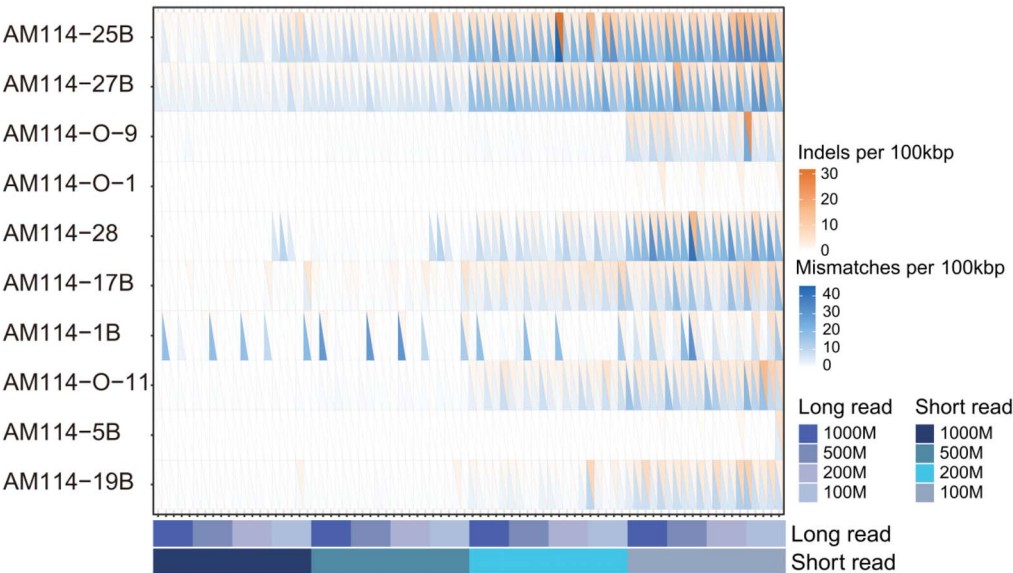

**Figure 9.** **The error rate of assemblies under different data volumes.**
The genomes were the same as in Figure 4. Mismatches per 100 kbp and indels per 100 kbp are denoted by blue and orange colors, respectively, in the heatmap.

## Impact of data volume on genome assembly accuracy

The integration of long-read data into the assembly process significantly enhances both the completeness and the rate of complete genome assembly. Concurrently, the accuracy of the assembled genomes remains an important consideration. Furthermore, we utilized the value of indels and mismatches calculated by QUAST (RRID:SCR_001228) [21] to evaluate the accuracy of genomes assembled from subsets of varying sizes, using the hybrid assembly from the original full dataset as the reference.

With 1,000 Mbp of short-read data, 76% of assemblies had mismatches below 1 per 100 kbp, and 97.5% had fewer than 10 per 100 kbp, with all assemblies exhibiting fewer than 10 indels per 100 kbp (Figure 9). When the short-read data was reduced to 500 Mbp, there was a slight decline in performance compared to 1,000 Mbp; however, 71% of assemblies still had mismatches below 1 per 100 kbp, 95% remained under 10 per 100 kbp, and all maintained fewer than 10 indels per 100 kbp, maintaining a high quality of assembly (see error rate table in GigaDB [20]). In contrast, reducing the short-read data volume to 200 Mbp or 100 Mbp led to a significant increase in the error rate for mismatches and indels per 100 kbp, with only 37% and 21% of assemblies, respectively, having mismatches under 1 per 100 kbp. Moreover, the average number of mismatches rose to 6.09 and 11.16 per 100 kbp. The above demonstrated that short-reads are particularly crucial for controlling the error rate.

## Feasibility of assembly for microbial communities

Metagenomics is an important application in the field of microbiology. To evaluate the performance of CycloneSEQ in assembling mixed microbial communities, we used the Gut Microbiome Standard, which includes 18 bacterial strains, two fungal strains, and one archaeal strain, for assessment. This sample was sequenced to generate clean data comprising 8.11 Gbp of long-read data and 11.79 Gbp of paired-end short-read data. The

N50 of the long-reads was 16.736 kbp, with an average length of 10.857 kbp. To evaluate the performance of short-read assembly, long-read assembly, and hybrid assembly, some commonly used assembly methods were used for sequence assembly. Moreover, metagenome-assembled genomes (MAGs) were produced by binning from short-read assembly (metaSPAdes [22]), long-read assembly (metaFlye [18]), and hybrid-assembly (Unicycler (RRID:SCR_024380), metaSPAdes, OPERA-MS [23]).

Under the same computing condition, hybrid assembly methods consume more time than short-read assembly by metaSPAdes and long-read assembly by metaFlye (Figure 10A). Unicycler was the most time-consuming method for hybrid assembly, and it required twice as much time as metaSPAdes and OPERA-MS. The long-read assembly produced fewer MAGs than the short-read assembly and the hybrid assembly (Figure 10B). Short-reads played a key role in improving the number of MAGs. The hybrid assemblies by Unicycler and metaSPAdes produced more single-contig MAGs (Figure 10C). The completeness of MAGs from the long-read assembly was lower than that of references and MAGs from short-read hybrid-assembly methods (Figures 10D, 11C). The contamination of MAGs from OPERA-MS (Figures 10E, 11D) and metaFlye (Figure 11D) was higher than that of references. The N50 of MAGs produced by the short-read assembly was shorter than the MAGs from long-read and hybrid-assembly methods (Figures 10F, 11E). The N50 of MAGs produced by OPERA-MS was only slightly higher than the N50 of MAGs by short-read assembly. In hybrid-assembly categories, metaSPAdes was the recommended tool for a hybrid assembly of short- and long-reads.

## DISCUSSION

As a new nanopore long-read sequencing platform, CycloneSEQ has demonstrated its sequencing performance and practical application in microbiology through this study. With an average read length of 11.7 kbp, comparable to other nanopore sequencing platforms, our analysis confirmed that this length is sufficient for assembling circular bacterial genomes. This performance is better than that of Oxford Nanopore Technologies when it was first launched, and it is comparable in terms of sequencing length [24, 25]. However, CycloneSEQ has notable deficiencies in base quality and remains behind the current Q20 performance of Pacific Biosciences HiFi and Oxford Nanopore Technology sequencing [26]. Therefore, integrating high-quality short-reads by DNBSEQ is a promising solution. Ultimately, we achieved a high-quality assembly with only 0.08 mismatches and 0.15 indels per 100 kbp compared to the reference, although this result may be influenced by strain variations during cultivation.

We evaluated the performance of CycloneSEQ and DNBSEQ on 10 bacterial strains. Sequencing and assembling genomes using only short-reads, only long-reads, and a hybrid of both, we found that hybrid assemblies consistently produced high-quality circular genomes, including potential small circular genomes from bacteriophages or plasmids. Our tests indicate that for some common species with GC content ranging from 36% to 60% and genome lengths between 2.17 Mbp and 6.41 Mbp, the hybrid assembly approach can successfully produce circular genomes. The hybrid approach of integrating DNBSEQ short-reads and CycloneSEQ long-reads combines their strengths effectively, producing complete and more accurate genome assemblies than either method alone. This improvement is particularly evident in the increased number of CDS, rRNA, and tRNA



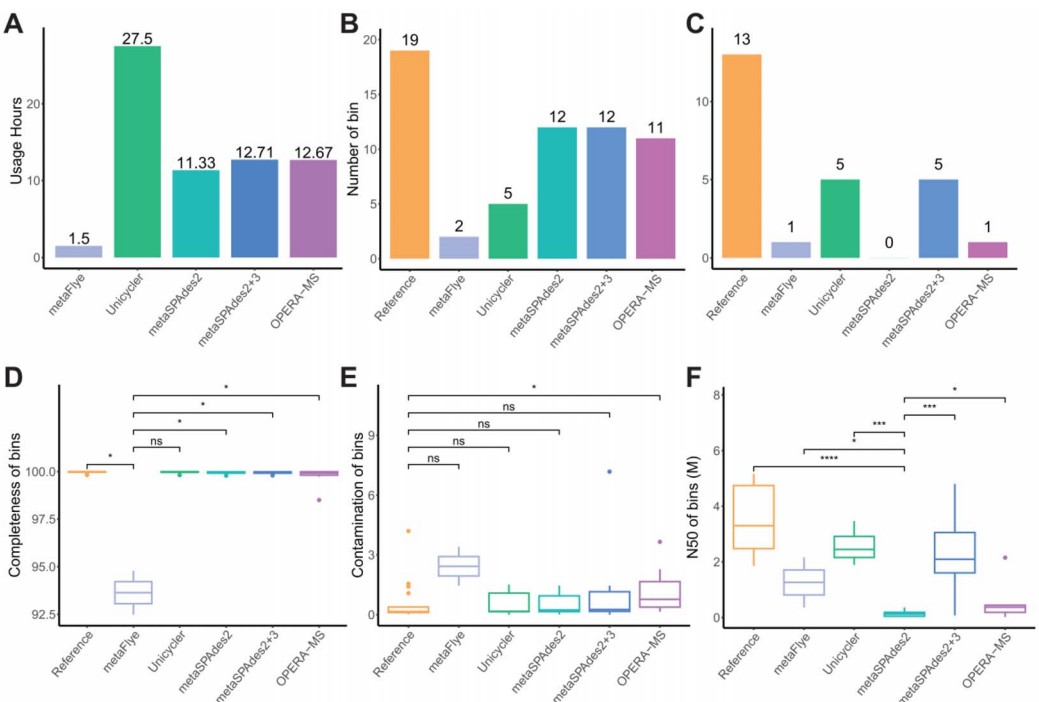

**Figure 10.** **High-quality MAGs from short-read assembly, long-read assembly, and hybrid assembly for mock metagenomic sequences.**
(A) The time consumption of sequence assembly. (B,C) Number of MAGs and single contig MAGs. (D–F) Completeness, contamination, and N50 of MAGs. MAGs: completeness ≥ 90%; contamination ≤ 10%. Long-read assembly: metaFlye; short-read assembly: metaSPAdes2; hybrid assembly approach: metaSPAdes2+3 and OPERA-MS. *, *p* value < 0.05; **, *p* value < 0.01; ***, *p* value < 0.001; ****, *p* value < 0.0001; ns, *p* value > 0.05.

coding genes in the complete genome compared to the draft. Due to limitations, we did not conduct sequencing tests on many more species or strains under more extreme conditions.

The ability to achieve high-quality assemblies with reduced long-read data volumes can make the hybrid assembly approach more cost-effective and accessible for various genomic research applications. This balance between data volume and assembly quality is crucial for optimizing resources in genomic studies. Using 1,000 Mbp of short-read data combined with varying amounts of long-read data, we achieved high success rates in assembling complete circular genomes, with up to 96% success. Even with reduced long-read data of 500 Mbp, the success rates remained robust. Meanwhile, the volume of short-read data significantly impacts assembly accuracy. With 1,000 Mbp of short-read data, 76% of assemblies had fewer than 1 bp mismatches per 100 kbp, and 97.5% had fewer than 10 bp mismatches per 100 kbp. Reducing short-read data to 500 Mbp slightly decreased the performance but maintained high accuracy. A further reduction to 200 Mbp or 100 Mbp led to a significant increase in error rates. These findings highlight that while long-read data is crucial for achieving complete genome assemblies, sufficient short-read data is essential for maintaining high accuracy. The hybrid assembly approach effectively balances these needs, making it the most efficient method for bacterial genome assembly. In conclusion, the hybrid assembly approach, particularly with adequate short-read data, is the optimal method for generating bacterial genome assemblies, balancing completeness, accuracy, and cost-effectiveness.

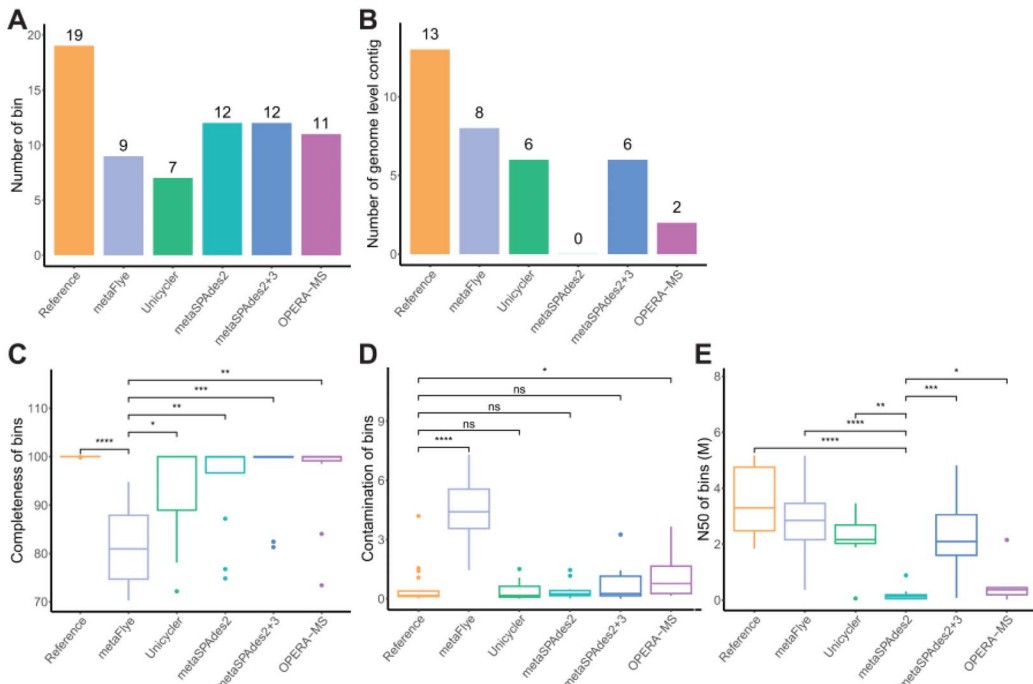

**Figure 11.   High-quality bins from short-read assembly, long-read assembly, and hybrid assembly for mock metagenomic sequences.**
(A) The number of bins. (B) Number of genome level contigs. (C–E) Completeness, contamination, and N50 of bins. Long-read assembly: metaFlye; short-read assembly: metaSPAdes2; hybrid assembly approach: metaSpAdes2+3 and OPERA-MS. *, $p$ value < 0.05; **, $p$ value < 0.01; ***, $p$ value < 0.001; ****, $p$ value < 0.0001; ns, $p$ value > 0.05.

Real microbial samples have complex microbial compositions and biochemical components, making metagenomic sequencing and assembly more challenging for nanopore-based sequencing platforms. To effectively evaluate the feasibility of CycloneSEQ in sequencing mixed microbial communities, we used the Gut Microbiome Standard as a substitute for metagenomic samples. Overall, using the metaSPAdes tool for the hybrid assembly of CycloneSEQ long-reads and DNBSEQ short-reads effectively combines the advantages of both short- and long-reads to produce complete and accurate genome assemblies. These findings report the utility of CycloneSEQ in metagenomics and highlight the advantages of hybrid assembly approaches. It is important to note that our study did not test real clinical samples, as the varying biochemical compositions between different samples could affect sequencing to different extents. This requires the design of more rigorous tests to achieve fair results. Future research should systematically test more real samples and focus on further optimizing the balance between short-read and long-read data to enhance assembly quality and efficiency. The CycloneSEQ long-read sequencing platform will facilitate these advancements in microbiome research.

## METHODS

The library construction and sequencing protocols used in this study are gathered in a protocols.io collection (Figure 12) [27].



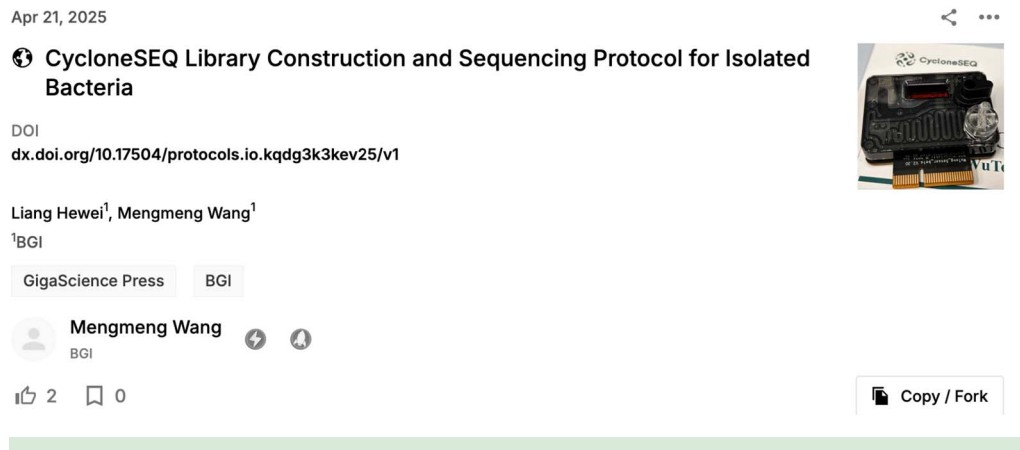

Apr 21, 2025

⊕ **CycloneSEQ Library Construction and Sequencing Protocol for Isolated Bacteria**

DOI
**dx.doi.org/10.17504/protocols.io.kqdg3k3kev25/v1**

Liang Hewei[1], Mengmeng Wang[1]
[1]BGI

GigaScience Press    BGI

Mengmeng Wang
BGI

👍 2    🔖 0    📋 Copy / Fork

**Figure 12.** A protocols.io Collection of protocols for CyloneSeq library construction and sequencing for isolated bacteria [27]. https://www.protocols.io/widgets/doi?uri=dx.doi.org/10.17504/protocols.io.kqdg3k3kev25/v1

## Sample collection, DNA extraction, library construction, and sequencing

A fecal sample was collected from a healthy man; the sample collection approved by the Institutional Review Board of BGI Ethical Clearance under number BGI-IRB 22112-T1. The sample was diluted and spread on agar culture mediums under anaerobic conditions. We then picked 114 single colonies and transferred each of them to 2 ml of liquid medium for further culture. We used 16S rDNA PCR to identify the species, and then we selected 10 diverse strains belonging to 9 species for further sequencing. The ATCC-BAA-835 DNA was extracted using the Qiagen QIAamp DNA Mini Kit for long DNA fragments, while the test strain DNA was extracted using the Magen MagPure DNA Kit for high-throughput applications. The CycloneSEQ library preparation and sequencing followed the manufacturer's guidelines and protocols [28]. Each sample, containing 2 μg of input DNA (≥21 ng/μL), was diluted with nuclease-free water to 192 μL, then mixed with 14 μL each of DNA repair buffers 1 and 2, 12 μL of DNA repair enzyme 1, and 8 μL of DNA repair enzyme 2. The mixtures were incubated in a thermocycler at 20 °C for 10 minutes, then at 65 °C for 10 minutes, and finally held at 4 °C. After incubation, the mixtures were purified with 1.0× DNA clean beads and eluted with 240 μL of nuclease-free water. The end-repaired samples were then mixed with 10 μL of sequencing adaptors, 100 μL of 4× ligation buffer, 40 μL of DNA ligase, and 10 μL of nuclease-free water, and incubated at 25 °C for 30 minutes. The ligated products were purified again with 1.0× DNA clean beads, resuspended with long fragment wash buffer, and recovered into 42 μL of elution buffer. The libraries were quantified using a Qubit fluorometer and sequenced on the CycloneSEQ WuTong02 platform according to sequencing protocols [29].

## Quality control and data evaluation

Long-read data was filtered using NanoFilt (RRID:SCR_016966) [30] with parameters "-q 10 -l 1000" to retain reads longer than 1,000 bp and with a quality score greater than Q10. Short-read data was processed using Fastp (RRID:SCR_016962) [31] with default parameters, except the length requirement was set to 50. The quality information of the data was evaluated using the tool seqtk (RRID:SCR_018927) [32], selecting the avgQ value from the

'fqchk' module as the average quality. The read lengths were extracted using a Python script [20], and a density plot was generated based on this information.

### Short-read, long-read, and hybrid assembly of the isolated genome

Short-read assembly was performed using Unicycler [16] with only the short reads '-1' and '-2' as input, and the '–depth_filter' set to 0.01 to remove low-depth contigs. For hybrid assembly, the same 'depth_filter' of 0.01 was used, with the addition of '-l' long reads as input, while all other parameters were set to default. For long-read assembly, we used Flye [33] with the filtered long reads as input using the '–nano-hq' option.

### Data splitting

Data splitting was performed using a custom Python script [20]. Based on the required data volume, the script divided the data into FASTQ files of different sizes. It is important to note that here, Mb represents 1,000,000 bases, not the 1,024-based system. Each read was treated as a unit, and the 'random.sample()' function was used for random selection. Reads were added one by one, and the total number of bases was calculated. When the addition of the last read met the required base count, the desired file was obtained. For paired short-reads, we assigned a sequence number to each read in the _1 and _2 files. Paired reads were then obtained by randomly selecting these sequence numbers.

### Completeness assessment and comparative evaluation of genome assemblies

The completeness of the genome was assessed using CheckM2 [34], while circularity was evaluated from the assembly results using Unicycler and Flye. We used QUAST [21] software for reference-based comparisons. To evaluate the assembly of the ATCC BAA-835 strain, we used the genome 'GCA_000020225.1' from GenBank as the reference. For the evaluation of actual samples, we used the hybrid assembly results from the complete dataset as the reference. Gene prediction and annotation were performed using Prokka (RRID:SCR_014732) [35], with coding sequences identified by Prodigal (RRID:SCR_011936) [36] and rRNA predicted using Barrnap (RRID:SCR_015995).

### Read mapping to genome and depth calculation

Bowtie2 (RRID:SCR_016368) [36] was used to map the short reads to the complete genomes with the '–very-sensitive' option. Samtools (RRID:SCR_002105) [38] was then used to convert the Bowtie2 output .bam file to the depth of each base site.

### Assembling, binning, annotation, and assessment for mock metagenomic data

SPAdes [22] (v3.15.5, -meta; RRID:SCR_000131) was used for short-read assembly. SPAdes (v3.15.5, -meta), OPERA-MS [23] (v0.9.0), and Unicycler [16] (v0.5.0, -l) were used to create a hybrid assembly of short-reads and long-reads. Flye [33] (v2.9.3, –meta –nano-raw) was used for long-reads assembling. For sequence assembly, 120 Gb memory and 24 threads were prepared. MAGs were constructed by Metawrap [39] (v1.3.2, -metabat2 -maxbin2 -concoct). Assembled genomes were annotated by GTDB-tk [19] (v2.3.2; RRID:SCR_019136). Completeness and contamination of MAGs were assessed by CheckM2 (v1.0.1), and genomic



quality assessments were conducted by QUAST [21] (v5.2.0). Average nucleotide identity between MAGs and reference genome of mock metagenome were calculated by FastANI [40] (v1.33; RRID:SCR_021091). R software (v4.1.1) was used for data analysis and data visualization.

## DATA AVAILABILITY
The data that support the findings of this study have been deposited into the CNGB Sequence Archive (CNSA) [41] of China National GeneBank DataBase (CNGBdb) [42] with accession number CNP0006129. Additional data is available in GigaDB [20].

## ABBREVIATIONS
CDS, coding sequence; MAG, metagenome-assembled genome.

## DECLARATIONS
### Ethics approval and consent to participate
A fecal sample was collected from a healthy man; the collection was approved by the Institutional Review Board of BGI Ethical Clearance under number BGI-IRB 22112-T1. The volunteer has signed the consent form (Version 2.0). All analyses were performed in accordance with the scope of the BGI-IRB 22112 research protocol.

### Competing interests
The CycloneSEQ was initially developed by BGI-Research and is now being marketed as an advanced technology. All the authors are employees of BGI-Research.

### Authors' contributions
Conceived and designed the study: YZou, LX, XX, HL, CL, XJ, WZ. Performed the analysis: HL, MW, TH, HW, WH, YW, LX, YJ, RG. Contributed reagents/materials/analysis tools: JC, FG, TZ, YD, YZhang, BW, XJ, XX. Wrote the paper: YZou, HL, MW. Supervised the work: LX, XX, YZou. All authors commented on the manuscript.

### Funding
This work was supported by grants from the Shenzhen Municipal Government of China (No. XMHT20220104017, CXB201108250097A, and KQTD20221101093603011).

### Acknowledgements
We also thank the colleagues at BGI-Shenzhen for sample collection and discussions, and China National GeneBank (CNGB) Shenzhen for DNA extraction, library construction, and sequencing.

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
