## [Editor Report]

Editor’s AssessmentWith the recent official launch of BGI’s new CycloneSEQ sequencing platform that delivers long-reads using novel nanpores, this paper presents benchmarking data and validation studies comparing short, long-rea data from other platforms and hybrid assemblies. This study tests the performance of the new platform in sequencing diverse microbial genomes, presenting raw and processed data to enable others to scrutinise and verify the work. Being openly peer-reviewed, and having scripts and protocols also shared for the first time helps provide transparency in this benchmarking process to increase trust in this new technology. On top of benchmarking typed strains, the technology also was tested with complex microbial communities. Yielding complete metagenome-assembled genomes (MAGs) which were not achieved by short- or long-read assemblies alone. By directly reading DNA molecules without fragmentation, the study demonstrating CycloneSEQ delivers long-read data with impressive length and accuracy, unlocking gaps that short-read technologies alone cannot bridge. Future work is expanding to real samples, with and fine-tuning the balance between short-read and long-read data for even faster, higher-quality assemblies.Editor’s AssessmentWith the recent official launch of BGI’s new CycloneSEQ sequencing platform that delivers long-reads using novel nanpores, this paper presents benchmarking data and validation studies comparing short, long-rea data from other platforms and hybrid assemblies. This study tests the performance of the new platform in sequencing diverse microbial genomes, presenting raw and processed data to enable others to scrutinise and verify the work. Being openly peer-reviewed, and having scripts and protocols also shared for the first time helps provide transparency in this benchmarking process to increase trust in this new technology. On top of benchmarking typed strains, the technology also was tested with complex microbial communities. Yielding complete metagenome-assembled genomes (MAGs) which were not achieved by short- or long-read assemblies alone. By directly reading DNA molecules without fragmentation, the study demonstrating CycloneSEQ delivers long-read data with impressive length and accuracy, unlocking gaps that short-read technologies alone cannot bridge. Future work is expanding to real samples, with and fine-tuning the balance between short-read and long-read data for even faster, higher-quality assemblies.

---

## [Reviewer Report]

Indicate in the comments box below whether you are happy with the changes made or if the manuscript is unacceptable.Comments on revised manuscriptThank you for the revisions to the manuscript. While many of my minor comments have been addressed, I still have concerns regarding my major comments, which have not been fully resolved. First, I appreciate that the data has now been made available on NCBI. However, the long-read datasets are labelled as Oxford Nanopore MinION data, which is misleading (example: SRR31850034). I understand this may be because SRA does not yet provide CycloneSEQ as a platform option, but this can be clarified through additional metadata. Specifically, the ‘design’ field for each SRA entry simply says ‘genome’, but it could have more detail, including that these are CycloneSEQ reads. The BioProject (PRJNA1194773) description could also include a clear statement that the long-read data is generated using CycloneSEQ. Second, I had requested a brief discussion of existing long-read platforms (ONT and PacBio) to provide context on where CycloneSEQ fits into the broader sequencing landscape. The authors have chosen not to include this, stating that they do not have direct comparison data. While I understand that such a comparison is not the purpose of this paper, I still believe that some mention of these platforms is necessary in the Background and/or Discussion sections. This paper introduces a new long-read technology for bacterial genome assembly, and readers will naturally want to understand how it relates to widely used alternatives. Finally, regarding my comment about supplementary figure labels, I still see the issue in the revised version provided for review. For example, the caption for Supplementary Figure S3 begins with ‘Supplementary Table S3.’ The authors stated that there were no errors, but this mislabelling remains in the PDF I received. As these concerns remain unresolved, I do not consider the manuscript acceptable in its current form.

---

## [Reviewer Report]

Reviewer name and names of any other individual's who aided in reviewerKeith RobisonDo you understand and agree to our policy of having open and named reviews, and having your review included with the published manuscript. (If no, please inform the editor that you cannot review this manuscript.)YesIs the language of sufficient quality?YesPlease add additional comments on language quality to clarify if neededIs there a clear statement of need explaining what problems the software is designed to solve and who the target audience is? YesAdditional CommentsIs the source code available, and has an appropriate Open Source Initiative license <a href="https://opensource.org/licenses" target="_blank">(https://opensource.org/licenses)</a> been assigned to the code?NoAdditional Commentsn/AAs Open Source Software are there guidelines on how to contribute, report issues or seek support on the code?YesAdditional CommentsN/A - no software presentedIs the code executable?Unable to testAdditional CommentsN/AIs installation/deployment sufficiently outlined in the paper and documentation, and does it proceed as outlined?Unable to testAdditional CommentsN/AIs the documentation provided clear and user friendly?NoAdditional CommentsN/AIs there enough clear information in the documentation to install, run and test this tool, including information on where to seek help if required?NoAdditional CommentsN/AIs there a clearly-stated list of dependencies, and is the core functionality of the software documented to a satisfactory level?NoAdditional CommentsHave any claims of performance been sufficiently tested and compared to other commonly-used packages? Not applicableAdditional CommentsIs test data available, either included with the submission or openly available via cited third party sources (e.g. accession numbers, data DOIs)?YesAdditional CommentsAre there (ideally real world) examples demonstrating use of the software? NoAdditional CommentsIs automated testing used or are there manual steps described so that the functionality of the software can be verified?NoAdditional CommentsAny Additional Overall Comments to the AuthorThis is a useful presentation of an emerging sequencing platform. Given the complex nature of nanopore signals and the difficulty of decoding them, it has been a pattern with the prior nanopore platform that improvements in basecalling software have yielded significant changes in basecalling performance. Therefore, it would be highly advantageous if the manuscript listed which specific versions / revision numbers of the basecalling software were used so that these results are properly contextualized for comparison to future results which may use newer basecalling software. Ideally, the publication would include a link to git (or similar) repository with the complete pipeline used to generate the resultsRecommendationMinor Revisions

---

## [Reviewer Report]

Reviewer name and names of any other individual's who aided in reviewerRyan WickDo you understand and agree to our policy of having open and named reviews, and having your review included with the published manuscript. (If no, please inform the editor that you cannot review this manuscript.)YesIs the language of sufficient quality?YesPlease add additional comments on language quality to clarify if neededIs there a clear statement of need explaining what problems the software is designed to solve and who the target audience is? YesAdditional Commentsn/aIs the source code available, and has an appropriate Open Source Initiative license <a href="https://opensource.org/licenses" target="_blank">(https://opensource.org/licenses)</a> been assigned to the code?YesAdditional Commentsn/aAs Open Source Software are there guidelines on how to contribute, report issues or seek support on the code?YesAdditional Commentsn/aIs the code executable?YesAdditional Commentsn/aIs installation/deployment sufficiently outlined in the paper and documentation, and does it proceed as outlined?YesAdditional Commentsn/aIs the documentation provided clear and user friendly?YesAdditional Commentsn/aIs there enough clear information in the documentation to install, run and test this tool, including information on where to seek help if required?YesAdditional Commentsn/aIs there a clearly-stated list of dependencies, and is the core functionality of the software documented to a satisfactory level?YesAdditional Commentsn/aHave any claims of performance been sufficiently tested and compared to other commonly-used packages? YesAdditional Commentsn/aIs test data available, either included with the submission or openly available via cited third party sources (e.g. accession numbers, data DOIs)?YesAdditional Commentsn/aAre there (ideally real world) examples demonstrating use of the software? YesAdditional Commentsn/aIs automated testing used or are there manual steps described so that the functionality of the software can be verified?YesAdditional Commentsn/aAny Additional Overall Comments to the AuthorThis paper was initially submitted as a Technical Release article, but it has since been changed to a Data Release article. The ReView site still lists the article as a Technical Release, and so the specific questions do not apply. As discussed with the editor, I have therefore put ‘n/a’ for each of the fields, and my entire review is here in the ‘Additional Overall Comments to the Author’ section. This manuscript introduces CycloneSEQ data as a means for producing complete bacterial genome assemblies, with a focus on hybrid assemblies made using a combination of CycloneSEQ data and DNBSEQ data. It also publicly provides deep CycloneSEQ+DNBSEQ read sets for a range of bacterial species. Major comments The reads for the project were made publicly available via CNGBdb (https://db.cngb.org/search/project/CNP0006129), but I found it to be unusably slow (both the HTTP website and the FTP data downloads). To ensure the data is accessible to a wide audience, I request that it also be hosted in another location to make it available to readers. For example, SRA, ENA or GigaDB. The paper makes no mention of the other major long-read platforms: Oxford Nanopore Technologies and Pacific Biosciences. Given the widespread use of these platforms (especially ONT) in bacterial genome assembly, some discussion on CycloneSEQ’s relative advantages or limitations would be beneficial. Minor comments Lines 100-103: this sentence (‘The GC content was sensitively affected…’) is not clear to me. How are the completeness and accuracy of the assembly affecting GC content? Figure S2 unnecessarily includes reference-vs-reference difference counts, which are by definition zero. Figure S2 could mention the genome (Akkermansia muciniphila ATCC BAA-835) in the caption – I did not immediately understand what 'for type strain' meant. I found Figure 5 difficult to read, with its use of colour to indicate accuracy. This data would be better shown using another visualisation (e.g. bar plot) that more clearly shows quantitative values. For the mixed microbial community analysis, it should be stated that Unicycler is exclusively designed for bacterial isolates (its documentation explicitly says to not use it on metagenomes). Some of the supplementary figures are erroneously labelled 'Supplementary Table'. Some stats on the metagenomic reads would be helpful: e.g. total bp for short and long reads, N50 for long reads, etc. The methods describe using seqtk, but the reference for this (#25) is SeqKit (a different tool), so either the tool in the methods or the reference is wrong.RecommendationMajor Revisions